# Optimization of nutritional strategies using a mechanistic computational model in prediabetes: Application to the J-DOIT1 study data

Julia H. Chen[1][¤], Momoko Fukasawa[2], Naoki Sakane[3]*, Akiko Suganuma[3], Hideshi Kuzuya[3], Shikhar Pandey[1], Paul D'Alessandro[1], Sai Phanindra Venkatapurapu[1], Gaurav Dwivedi[1]

**1** PricewaterhouseCoopers LLP, Pittsburgh, PA, United States of America, **2** PwC Consulting LLC, Chiyoda-ku, Tokyo, Japan, **3** Division of Preventive Medicine, Clinical Research Institute, National Hospital Organization Kyoto Medical Center, Kyoto, Japan

¤ Current address: COTA Healthcare, New York, NY, United States of America

* nsakane@gf6.so-net.ne.jp

**Data Availability Statement:** Individual patient data in this study was obtained from the JDOIT-1 Study with permission from the study group.

## Abstract

Lifestyle interventions have been shown to prevent or delay the onset of diabetes; however, inter-individual variability in responses to such interventions makes lifestyle recommendations challenging. We analyzed the Japan Diabetes Outcome Intervention Trial-1 (J-DOIT1) study data using a previously published mechanistic simulation model of type 2 diabetes onset and progression to understand the causes of inter-individual variability and to optimize dietary intervention strategies at an individual level. J-DOIT1, a large-scale lifestyle intervention study, involved 2607 subjects with a 4.2-year median follow-up period. We selected 112 individuals from the J-DOIT1 study and calibrated the mechanistic model to each participant's body weight and HbA1c time courses. We evaluated the relationship of physiological (e.g., insulin sensitivity) and lifestyle (e.g., dietary intake) parameters with variability in outcome. Finally, we used simulation analyses to predict individually optimized diets for weight reduction. The model predicted individual body weight and HbA1c time courses with a mean (±SD) prediction error of 1.0 kg (±1.2) and 0.14% (±0.18), respectively. Individuals with the most and least improved biomarkers showed no significant differences in model-estimated energy balance. A wide range of weight changes was observed for similar model-estimated caloric changes, indicating that caloric balance alone may not be a good predictor of body weight. The model suggests that a set of optimal diets exists to achieve a defined weight reduction, and this set of diets is unique to each individual. Our diabetes model can simulate changes in body weight and glycemic control as a result of lifestyle interventions. Moreover, this model could help dieticians and physicians to optimize personalized nutritional strategies according to their patients' goals.

Queries related to the patient data should be directed to the JDOIT-1 study group (nsakane@gf6.so-net.ne.jp). The simulation model used in this study was previously published: https://doi.org/10.1371/journal.pone.0192472.

**Funding:** This work was supported by JSPS KAKENHI Grant Number 18k01988.The funders had no role in study design, data collection and analysis, decision to publish, or preparation of the manuscript. PricewaterhouseCoopers, LLP provided support in the form of salaries for the following authors - [JHC, MF, SP, PMD, SPV, GD] but did not have any additional role in the study design, data collection and analysis, decision to publish, or preparation of the manuscript.

**Competing interests:** The authors have declared that no competing interests exist.

## Introduction

In the National Diabetes Statistics Report 2020 [1] from the Centers for Disease Control and Prevention (CDC), it was estimated that about 34.2 million people (~10.5% of the US population) are diabetic, accounting for $237 billion in direct medical expenses and $90 billion in indirect medical costs. Globally, diabetes is now considered an epidemic, affecting more than 420 million individuals (~6% of the world's population) [2] and can lead to various complications [3]. Although lifestyle factors, such as diet composition, exercise, and sleep play an important role in type 2 diabetes (T2D) development [4–6], the response to similar lifestyle changes varies dramatically among individuals [7]. This inter-individual variability could be due to pathophysiological differences among individuals [8], differences in the physiological response to dietary or exercise intervention [9], and other factors [7]. Therefore, it is desirable to develop a framework for designing individualized strategies to achieve defined health goals targeted toward preventing or delaying the onset of diabetes. However, a limited understanding of the causes of inter-individual variability makes it challenging to design individualized interventions, e.g., diet plans, for diabetes prevention.

Precision nutrition aims to prevent and manage chronic diseases by tailoring dietary interventions or recommendations considering the individual's genetic background, metabolic profile, gut microbiome, and environmental exposure. Currently, the field of precision nutrition is faced with challenges such as the high cost of genomics and metabolomics technologies and lacks robust and reproducible results in studies on precision nutrition [10, 11]. In contrast to precision nutrition, there are general strategies that do not attempt to individualize dietary recommendations, such as low-carbohydrate or low-fat diets. Several studies have shown the effectiveness of both low-fat and low-carbohydrate diets for weight control and reduction of cardiovascular risk [12–16]. The US Diabetes Prevention Program (DPP) [4] and Finnish Diabetes Prevention Study (DPS) [5] on lifestyle modifications (low-fat diets, lifestyle changes targeting 5–7% weight loss, and exercise habits) have demonstrated a reduction in the burden of T2D by up to 58% [4]. A meta-analysis [17] of data from 11 randomized controlled studies (1369 participants) revealed that a low-carbohydrate diet can aid in weight reduction [18]. Moreover, a low-carbohydrate diet was also found to be more effective in glycemic control compared to a low-fat diet in patients with T2D [19].

While generalized dietary strategies such as low-fat and low-carbohydrate diets have been successful to varying degrees in various contexts, it is unclear whether and which approach may be successful for a specific individual. Advances in precision nutrition are promising but still under development and may not be cost-effective [10]. To address the need for individualized dietary recommendations, we explore the use of a computational simulation modeling tool in this work.

We previously developed a computational simulation model [20] of macronutrient metabolism and T2D onset and progression and tested it using data from DPP. The impact of lifestyle changes on endpoints including body weight and HbA1c were predicted at the individual level over a period of 3 years for 315 subjects from the DPP study. The mean prediction error for individual-level body weight and HbA1c changes over the three-year period was approximately 5% each. This suggests that the model can be used to predict and optimize individual-level responses to lifestyle changes. To our knowledge, currently there are no studies on the optimization of dietary strategies for preventing T2D using simulation modeling based on physiological principles.

The Japan Diabetes Outcome Intervention Trial-1 (J-DOIT1), a nationwide pragmatic cluster-randomized controlled trial, showed that participants who received telephone calls more frequently had a significantly reduced risk (41%) of T2D development [21, 22]. Herein, using

the simulation model, we simulated and analyzed data from J-DOIT1 [22] to evaluate factors affecting inter-individual variability in response to diet change, including endogenous (physiological characteristics) and exogenous (e.g., macronutrient intake) factors. The model adequately described individual-level body weight and HbA1c dynamics over time as observed in J-DOIT1. We also demonstrate how the simulation approach may be used to optimize diet therapy for individuals to achieve specific health goals.

## Materials and methods

### Simulation model

A previously developed computational simulation model of T2D was used [20]. This computational simulation model of T2D, referred to as the "model" henceforth, is based on the physiological mechanisms underlying the onset and progression of T2D. Important physiological (endogenous) and lifestyle (exogenous) factors involved in T2D are represented in the model. Exogenous factors influencing T2D are represented through dietary intake of macronutrients, i.e., carbohydrates, fats, and proteins, as well as energy expenditure through physical activity. Endogenous or physiological drivers of T2D are represented mechanistically in the model through physiological processes occurring at the cellular, tissue, and whole-body levels.

At a high level, the model mathematically represents the dynamics of dietary intake of carbohydrates, fats, and proteins, their breakdown and transportation into major tissue compartments through the bloodstream, and the interconversion of metabolic species into stored and active forms (Fig 1). A module representing the pancreas regulates insulin secretion into the bloodstream. Cellular processes modulating the activation of insulin receptors by insulin drive the development of insulin resistance, which in turn controls several processes, including glucose uptake by tissues. Oxidation of macronutrients generates ATP, which provides energy for basal metabolism and physical activity. Changes in caloric intake, macronutrient composition, and/or physical activity levels have a cascading impact on all components of the model, leading to changes in key outputs, such as body weight, plasma glucose, and HbA1c.

Details regarding the development and validation of the model have been described previously [20]. For the analysis presented here, the model described in the original publication was used.

### Digital twins

The computational simulation model comprises several numerical parameters that can be adjusted to fit model outputs, such as body weight and HbA1c trends over time, to the observed data of a specific individual. A model that has been calibrated to represent the historical data of a specific individual can be considered a "digital twin" of the individual. The digital twin can be used to simulate experiments with various lifestyle modifications quickly and safely in a virtual *in silico* environment. The model's ability to use digital twins to predict body weight and HbA1c was previously tested using individual-level data from DPP [4, 20]. The concept of digital twins was applied in the work presented here. Digital twins were created for individuals selected from the J-DOIT1 study by calibrating instances of the model using a previously described method [20]. The digital twins were then used to simulate various scenarios, e.g., effect of variation in fat and carbohydrate composition of diet, to understand and analyze the variability in individual responses to interventions.

### J-DOIT1 study

The Japan Diabetes Outcome Intervention Trial-1 (J-DOIT1) is a pragmatic, cluster-randomized, controlled trial conducted in Japan. The trial investigated the impact of lifestyle coaching

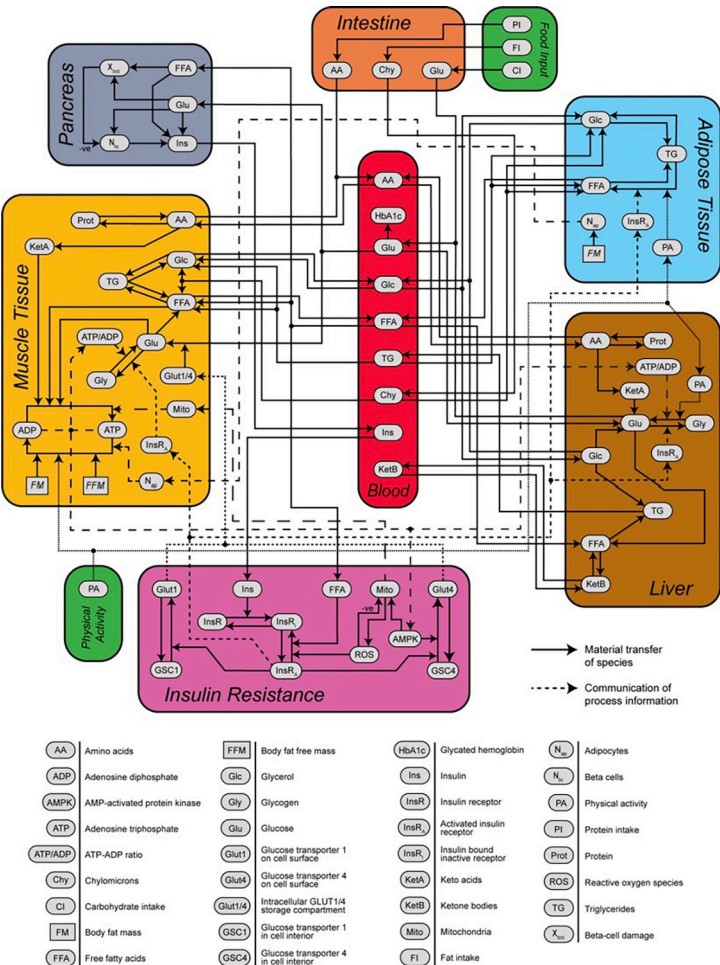

**Fig 1. Simulation model.** Schematic representing a previously developed and validated mechanistic model of diabetes onset and progression. (Adapted from [20]).

delivered through telephone calls on T2D development in high-risk individuals in a primary healthcare setting [21]. A total of 2607 individuals (1240 in the intervention arm and 1367 in the control (placebo) arm) completed the study with a median follow-up period of 4.2 years [22]. Participants in the intervention arm received lifestyle support telephone calls from healthcare providers over a 1-year period. The intervention arm was further divided into three lifestyle support centers designated as centers A, B, and C. During the 1-year period for which telephone-delivered lifestyle support was provided, participants in centers A, B, and C received 3, 6, and 10 support calls, respectively. Thus, centers A, B, and C can be considered as low-, medium-, and high-support call frequency groups, respectively. The control arm did not receive any support through telephone but received periodic newsletters on diabetes and diabetes prevention. The participants were followed-up annually. The onset of T2D status was assessed as the primary outcome, and the other outcomes included body weight and HbA1c. The detailed study design, including patient recruitment, inclusion/exclusion criteria, participants' consent and ethics committee approval of J-DOIT1 study can be found in the original study article [21].

## Selection of the analysis dataset

A total of 112 unique J-DOIT study participants were selected for the individual-level analysis using the following algorithm (Fig 2). For each subject in the J-DOIT1 dataset, the percentage change in the body weight and HbA1c level from baseline to the end of the intervention was calculated. The degree of response for each subject was defined as the sum of the percentage decrease in the body weight and HbA1c. Individuals with the largest collective decrease in body weight and HbA1c were considered as the "best responders" while those with the least decrease or greatest increase were considered as the "worst responders." Using this definition, 29 best responders were selected from the intervention arm, with 10 each drawn from the low- and high-support call frequency groups, and 9 from the medium-support call frequency group (corresponding to centers A, C, and B, respectively, as described above). Similarly, 30 worst responders were selected from the intervention arm, with 10 each from the low-, medium-, and high-support call frequency groups. Thus, 59 subjects were selected from the intervention arm with nearly equal representation of the best and worst responders from all three call frequency groups.

Subsequently, a baseline-matched subject from the control arm was identified for each of the 59 subjects selected from the intervention arm. The method used to identify baseline-matched subjects is described next. Sex, height, baseline age, baseline body weight, and baseline HbA1c levels of each subject from the intervention arm were selected as the reference values. Matched subjects in the control arm with the same sex, height within ±3 cm, baseline age within ±2 years, baseline body weight within ±4 kg, and baseline HbA1c within ±0.3% of the reference value were selected. Of the subjects from the control arm that matched these criteria, the subject with the smallest difference in body weight and HbA1c level was selected as the baseline-matched pair of the intervention subject. If a matched subject from the control arm could not be found for a subject from the intervention arm, the intervention arm subject was dropped and another intervention subject was selected.

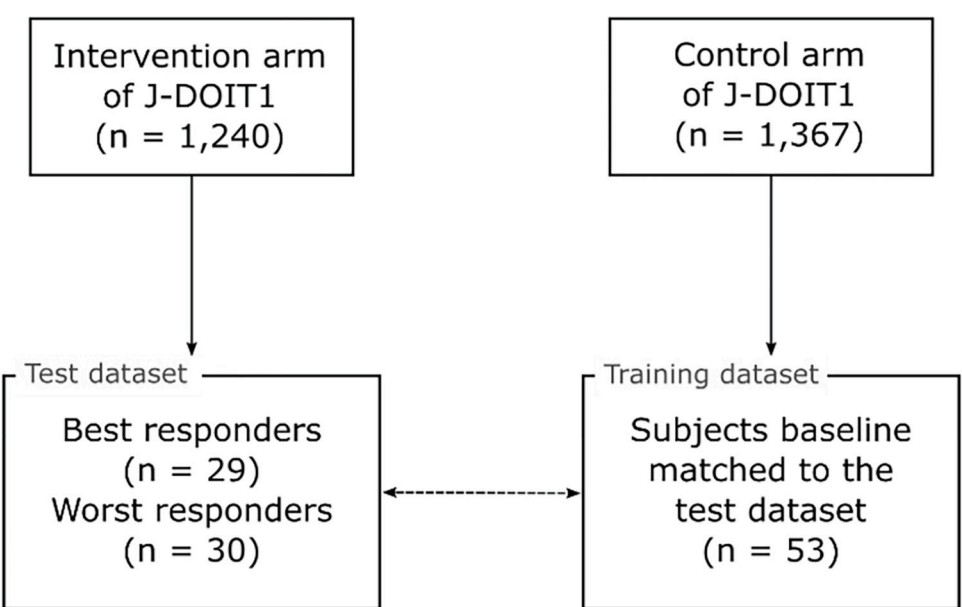

**Fig 2. Subject selection and study design.** 112 unique subjects were selected for individual-level analysis. 59 subjects were selected from the intervention arm of J-DOIT1 with a nearly equal distribution over three call frequency groups and two response categories within each call frequency group. 53 subjects from the control arm were found to be the best baseline-matched pairs of the 59 subjects from the intervention arm.

Using these criteria, 53 unique subjects were selected from the control arm. The number of unique subjects selected from the control arm was less than 59 because 6 control subjects were baseline-matched to 2 intervention subjects each. The 53 matched subjects from the control arm were used as the training dataset, and the other 59 from the intervention arm were used to test the model predictions. Further details of the training and test processes are described below.

## Model calibration and testing

The model consists of two types of parameters: 1) physiological parameters or parameters representing endogenous processes that are inherent to the individual and do not change over the course of the simulation; and 2) lifestyle parameters, which can change dynamically over time because of interventions.

**Calibration of the training dataset.** For the training dataset, a subset of physiological parameters was calibrated in addition to lifestyle parameters (Table 1) to fit the model's predicted body weight and HbA1c levels to each subject's measured body weight and HbA1c time course over the duration of the J-DOIT1 study. While the physiological parameters were constant for an individual by design, step changes in lifestyle were allowed at discrete time points over the duration of the simulation. The simulation was stopped at discrete time points, lifestyle parameters were changed, and the simulation continued using the final state of the last segment as the initial state of the new segment. The discrete time points when the simulation start-stop occurs were determined empirically by manually observing the trends in body weight and HbA1c. Whenever a previously decreasing trend in either body weight or HbA1c was followed by an increasing trend or vice-versa, a lifestyle change was introduced, assuming that such changes in body weight or HbA1c could only be driven by lifestyle factors. An effort was made to explain the entire trajectory of body weight and HbA1c with the minimum number of discrete lifestyle changes. A maximum of 4 such step changes to lifestyle were permitted over the entire follow-up period of approximately 4 years. The set of physiological and lifestyle parameters that resulted in the best achievable fit to the measured body weight and HbA1c

**Table 1. Model parameters calibrated to fit individual subject time-course data.**

| Category | Parameter fit to individual subject | Parameter symbol[3] |
|---|---|---|
| Physiology parameters[1] | Basal carbohydrate intake requirement to maintain steady state body weight | $CI_0$ |
| | Basal fat intake requirement to maintain steady state body weight | $FI_0$ |
| | Maximal HbA1c concentration | $Cmax_{hba1c}^{BLD}$ |
| | Initial HbA1c concentration | $C_{hba1c}^{BLD}(t=0)$ |
| | Maximal inhibitory effect of free fatty acids (FFA) on insulin signaling | $\alpha_{dep\_ffa}$ |
| | FFA concentration for half maximal inhibition of insulin signaling | $k_{dep\_ffa}$ |
| | Extent of pancreatic beta cell damage due to glucotoxicity, lipotoxicity, and inflammation | $\alpha_{bc,s\_ros}$ |
| Historical lifestyle parameters[1] | Carbohydrate intake prior to study start | $CI$ |
| | Fat intake prior to study start | $FI$ |
| Lifestyle parameters during the study[2] | Carbohydrate intake at various time points during the study | $CI_1,CI_2,\ldots,CI_4$ |
| | Fat intake at various time points during the study | $FI_1,FI_2,\ldots,FI_4$ |
| | Change in physical activity at various time points during the study | $\Delta PA_1,\ldots,\Delta PA_4$ |
| | Time points at which carbohydrate intake, fat intake, and physical activity change during the study | $T_1,T_2,\ldots,T_4$ |

[1]Calibrated only for baseline-matched subjects from the control group, i.e., training dataset.

[2]Calibrated for all subjects.

[3]Symbols as used in the original model [20].

time course of an individual was accepted as the parameter set for that individual. As a result of this process, each subject from the training set had a unique combination of physiological and lifestyle parameters that defined the digital twin of that subject.

**Calibration of the test dataset.** As described above, the training dataset was obtained by baseline-matching the test data. The baseline-matched pairs comprising one subject each from the training and test datasets were of the same age and sex and had similar body weight, height, body mass index, and HbA1c at baseline. Because of this similarity in their baseline attributes, we assumed that the physiological parameters, as well as carbohydrate and fat intake prior to the start of the study were identical for both subjects in a baseline-matched pair. The implication of this assumption is that the physiological parameters of each test subject are predetermined by their corresponding match from the training dataset; any differences in the observed body weight and HbA1c time courses of the pair during the J-DOIT1 study could be explained only by differences in their lifestyles, such as carbohydrate and fat intake and exercise changes during the study. This limits the range of responses that can be achieved for individuals in the test dataset because lifestyle is the only variable input to the model and serves as a mechanism to test the model's ability to forecast individual responses. For the test dataset, only step changes in the category "Lifestyle parameters during the study" (Table 1) were allowed. The time points at which these step changes in lifestyle were introduced in the simulation were determined, as explained above in "Calibration of the training dataset" section, empirically based on trends in body weight and HbA1c. Changes in lifestyle parameters were calibrated for each test subject to determine the best fit to individual time courses of body weight and duration over the duration of the J-DOIT1 study. Across all training and test subjects, the median (range) of the number of discrete lifestyle changes required to fit the body weight and HbA1c time courses for each subject was 2.5 (1–4). 10 subjects required only 1 lifestyle change, 46 required 2, 49 required 3, and 7 subjects had 4 lifestyle changes. A median (range) of 6 (4–6) body weight measurements and 6 (3–6) HbA1c measurements were available for each subject to calibrate the lifestyle change parameters. More than 75% of all subjects in the training and test datasets had 6 measurements each for body weight and HbA1c.

Parameter calibrations were performed using the differential evolution algorithm [23] and the objective function to be minimized was the sum of the squared errors over all time points for body weight and HbA1c. As described in the "Calibration of the training dataset" section, up to 4 lifestyle changes were applied to each subject in discrete segment over the entire follow-up period. During the fitting process, all lifestyle changes for a particular subject were allowed to change simultaneously rather than fitting each segment separately.

For calibration, each data point was assumed to have an inherent measurement error, and the objective function was designed to consider this error. Body weight was assumed to carry a measurement error of ±1 kg based on previous studies on imprecision in the measurement of body weight using weighing scales [24, 25]. HbA1c was assumed to have a measurement error of ±0.15 percentage points, which is approximately 3% of the median HbA1c value of 5.5% across all data points in this analysis. A 3% error is well within the ±5% measurement error considered acceptable by the National Glycohemoglobin Standardization Program (NGSP) [26]. Based on a study of Japanese individuals, the measurement error for HbA1c was estimated to be 0.17 percentage points [27].

The following objective function was used for parameter estimation for each subject:

$$\Phi(\theta) = \sum_i \sum_j \frac{(y_{ij}(\theta) - x_{ij})^2}{e_i^2}$$

where $\theta$ represents the model parameter vector, $i$ is either body weight or HbA1c and $j$

represents all the time points at which biomarker $i$ is measured for the subject. $y_{ij}$ is the model simulation output of biomarker $i$ at time $j$ whereas $x_{ij}$ is the observed value. $e_i$ is the measurement error associated with biomarker $i$, such that $e_{body\ weight} = 1\ kg$ and $e_{HbA1c} = 0.15\%$.

## Simulations

To test the effects of dietary changes and determine the optimal diet, simulations were performed using the calibrated digital twins of the study subjects. Starting from the baseline age (age at the start of J-DOIT1) of a digital twin, a random step change in carbohydrate and fat intake was introduced. Keeping all other parameters constant, the body weight and HbA1c time-courses were simulated with diet change. This process was repeated 2000 times for each digital twin using a Monte Carlo approach with macronutrient changes sampled from a uniform random distribution in the range baseline value– 25% to baseline value + 25%. The 2000 simulations and parameter samples were independent of each other. As only 3 parameters were sampled (change in carbohydrate, fat, and protein intake), a sample size of 2000 was considered appropriate to reasonably cover the parameter space along all three dimensions within the ±25% cube. Simulation outputs were recorded and analyzed.

## Results

### The model successfully captures individual-level dynamics of body weight and HbA1c

The model was fit to individual time-courses of body weight and HbA1c by calibrating both physiological and lifestyle parameters (Table 1) for the training dataset and only lifestyle parameters for the test dataset, as described in the Materials and methods section. Results showed that individual-level changes in the body weight and HbA1c over time were captured well by the model for both the training and test datasets (Fig 3, S1–S6 Figs). Fig 3 is a selected example of model training and test results. Prediction accuracy varies by subject and time point, and in some cases, it is larger than that represented in Fig 3. In one extreme example, the body weight of subject Test-005 was overpredicted by nearly 10 kg at the last time point occurring nearly 5 years from baseline (S1 Fig). In another instance, serum HbA1c was underpredicted by nearly 0.5 percentage points for subject Test-029 at the last measurement (S2 Fig). For a comprehensive visual comparison of the predicted values with the measured values across all time points, refer to S1-S7 Figs. Despite these extreme examples, overall the model performs well at predicting the measured values. The prediction error (mean [±SD]) across all data points in the training dataset for body weight was 0.7 kg (±0.8) and for HbA1c it was 0.08% (±0.08). In terms of percentage error (mean [±SD]), body weight of subjects in the test dataset was predicted with an error of 1.1% (±1.0) and HbA1c with an error of 1.4% (±1.4) relative to the actual measurement (Table 2).

### Changes in caloric balance alone do not fully explain the variability in individual response

After calibration and testing against individual time-course data, the model was used to estimate the likely caloric change per individual that led to the observed change in body weight. Calibrated digital twins were used to estimate the caloric change for each individual due to modifications in diet and exercise during the period between baseline and first follow-up in the intervention period of the J-DOIT1 study (median duration 1 year). The total caloric change (decrease or increase) was defined as the sum of changes in caloric intake due to diet change and caloric expenditure due to exercise. Changes in daily calories from baseline to the

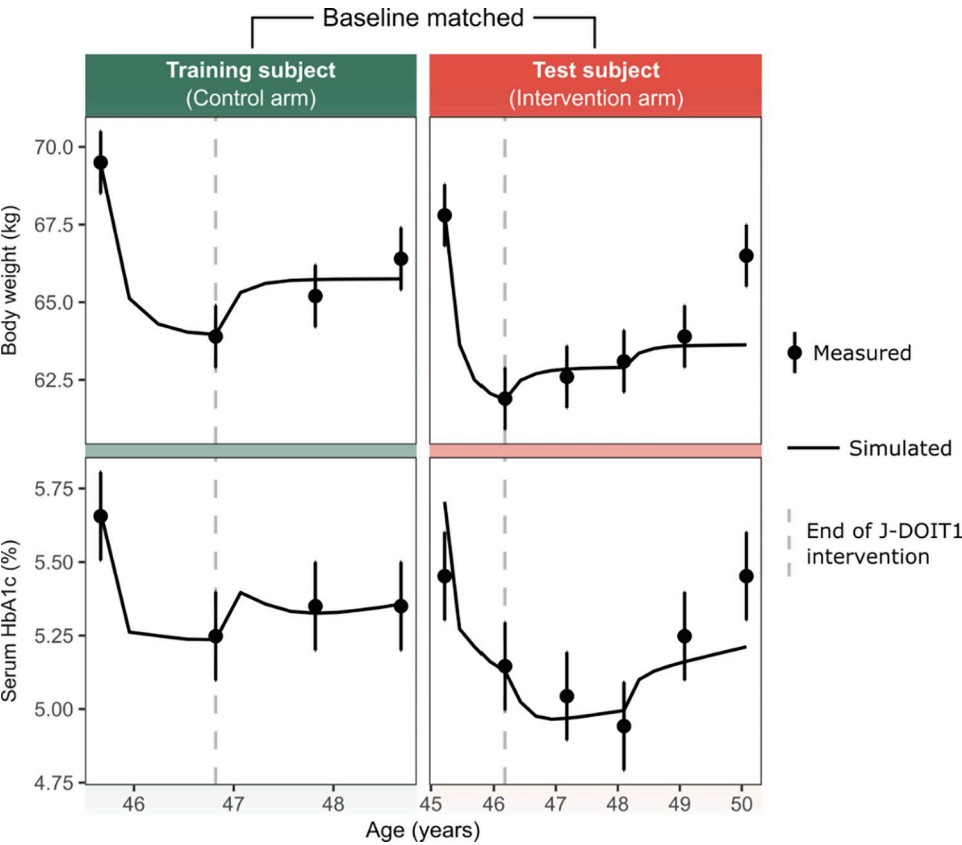

**Fig 3. An example of model prediction for a pair of baseline-matched training and test subjects.** Panels on the left-hand side represent a subject from the training data set. Panels on the right-hand side show the baseline-matched subject from the test data set. The test subject is from the high call frequency group and was classified in the best responder category. The error bars around the measured values are assumed measurement errors, ±1 kg for body weight and ±0.15 points for HbA1c, as described under model calibration in the Materials and methods section.

first post-baseline follow-up were estimated for each individual using the calibrated model parameters. The measured change in body weight during the same interval (baseline to the first follow-up) was also calculated. The model-estimated caloric change versus the observed weight change from baseline to the first follow-up is shown in Fig 4. The measured change in body weight generally increased with the model-predicted increase in caloric intake, with a Pearson correlation coefficient of 0.82 (Fig 4). The model predicted that similar caloric changes could lead to a wide range of responses in terms of body weight changes across individuals, as indicated by the spread of the points along the y-axis in Fig 4. When a linear regression model was fitted to the data (solid gray line in Fig 4), the residual error ranged from -4.6 kg to +7.0 kg with a residual standard error of 2.5 kg, indicating a relatively wide spread of body weights around the line of best fit. Similar trends were observed for HbA1c levels (S8 Fig).

We also explored the question of whether the degree of response (change in body weight and HbA1c) could be related to endogenous characteristics (physiology parameters defined in Table 1) of subjects. Our hypothesis was that certain ranges or values or combinations of the physiological model parameters (model parameters other than lifestyle factors) could make weight loss easier, and that such trends would be observable through correlation of certain parameters with the response. None of the calibrated physiology parameters, either alone or in

**Table 2. Model prediction errors.** Prediction errors are shown after grouping subjects using various criteria.

| Group | Number of unique subjects | Biomarker | Absolute prediction error Mean (±SD) | Percentage prediction error Mean (±SD) [% of measured] |
|---|---|---|---|---|
| All subjects | 112 | Body weight | 1.0 kg (±1.2) | 1.5 (±1.6) |
| | | HbA1c | 0.14% (±0.18) | 2.5 (±3.4) |
| Control (Training data) | 53 | Body weight | 0.7 kg (±0.8) | 1.1 (±1.0) |
| | | HbA1c | 0.08% (±0.08) | 1.4 (±1.4) |
| Intervention (Training data) | 59 | Body weight | 1.3 kg (±1.4) | 1.8 (±1.9) |
| | | HbA1c | 0.18% (±0.23) | 3.4 (±4.2) |
| Best responders | 29 | Body weight | 1.6 kg (±1.8) | 2.4 (±2.3) |
| | | HbA1c | 0.20% (±0.26) | 3.6 (±4.9) |
| Worst responders | 30 | Body weight | 1.0 kg (±0.9) | 1.4 (±1.2) |
| | | HbA1c | 0.17% (±0.19) | 3.1 (±3.4) |

The prediction error (mean [±SD]) across all data points in the test dataset for body weight was 1.3 kg (±1.4), and for HbA1c it was 0.18% (±0.23). In terms of percentage error, body weight was predicted for the test dataset with an error of 1.8% (±1.9) and HbA1c with an error of 3.4% (±4.2) relative to the measured value (Table 2).

linear combinations, were found to be correlated with changes in body weight or HbA1c. Similar hypotheses have been tested in clinical studies [28] and corroborate our modeling analysis.

## Diet therapy is predicted to have maximal effectiveness when optimized individually

Simulations were performed to determine the "optimal" diet for achieving a 5–7% reduction in body weight over a period corresponding to the duration between baseline and 1-year post-intervention. Digital twins of the J-DOIT1 study subjects from the test dataset (N = 59) were simulated with various random modifications to their carbohydrate, fat, and protein intake. Each macronutrient was sampled from a uniform distribution within ±25% of its baseline value for the digital twin, which resulted in a total caloric change distribution spanning the range ±25% (S9 Fig). Further details of the method are provided under the sub-heading Simulations of the Materials and methods section. Diets that led to a 5–7% reduction in body weight were selected as optimal diets. Using this approach, optimal diets could be identified for 48 of the 59 subjects; the remaining 11 subjects probably needed diet changes beyond the ±25% range simulated. Of the 48 subjects for whom optimal diets could be identified, one subject had only a single diet change within the sampled range of ±25% (24% reduction in carbohydrate and 25% reduction in fat intake) that led to a 5–7% reduction in body weight. For all other subjects (N = 47), sets of various diet compositions, as opposed to a single optimal diet, led to the target weight reduction of 5–7% (a range of 3 to 668 diet compositions for each subject with a median of 186 diet compositions). Furthermore, this set of diets was unique to each participant. A comparison of the carbohydrate and fat distributions of the optimal diets for two subjects is shown as an example in Fig 5. Changes in protein intake were not correlated with weight changes (S10 Fig). This is consistent with previous studies on the effects of changes in protein intake [29], which suggest that altering protein intake by itself does not significantly affect body weight. Given the limited impact of protein change, further analysis was limited to carbohydrates and fats.

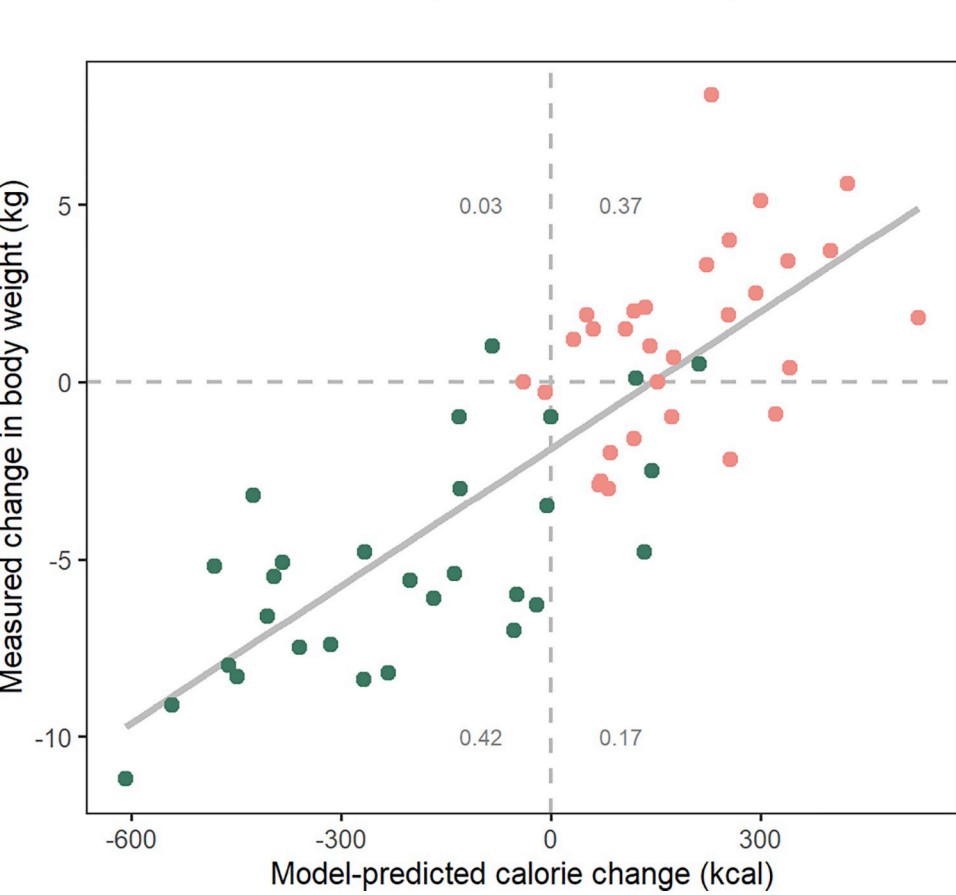

**Fig 4. Model-predicted caloric change versus weight change for subjects in the intervention arm.** The measured change in body weight from baseline to the first follow-up during the J-DOIT1 intervention (median duration 1 year) is plotted on the y-axis for subjects in the intervention arm. The x-axis shows model-estimated change in calories per day due to both diet and exercise changes averaged over the same period. The gray number in each quadrant is the fraction of data points in that quadrant. The data points fit a linear regression model (solid gray line) with $r^2 = 0.67$ and a residual standard error of 2.5 kg, indicating a relatively wide spread around the line of best fit. Best and worst responders were defined based on the total percent change in body weight and HbA1c one year after the end of the J-DOIT1 intervention.

The two subjects presented in Fig 5 show qualitatively different distributions of optimal diet changes. For subject ID Test-041, carbohydrate intake could change over a wide range of approximately -25% to +25% but fat change needed to be more narrowly restricted between approximately -25% to -10%. Contrary to this, for subject ID Test-044, fat change could range between -25% to +25% but carbohydrate change had to be restricted to a narrower range (-25% to -5%). In an alternative interpretation, subject ID Test-041 is predicted to be more sensitive to fat change than to carbohydrate change and should more precisely control fat intake to achieve the targeted weight loss. Subject ID Test-044, on the other hand, is predicted to be more sensitive to carbohydrate change; this subject should pay more attention to regulating carbohydrate intake but can be less particular about controlling fat intake.

In addition to the 5–7% body weight reduction for subject Test-041 (Fig 5), an additional target of 0.1–0.2 point reduction in HbA1c was added. Applying this additional target led to further refinement of the optimal diets and a subset of the original optimal diets was predicted to simultaneously achieve both targets (Fig 6).

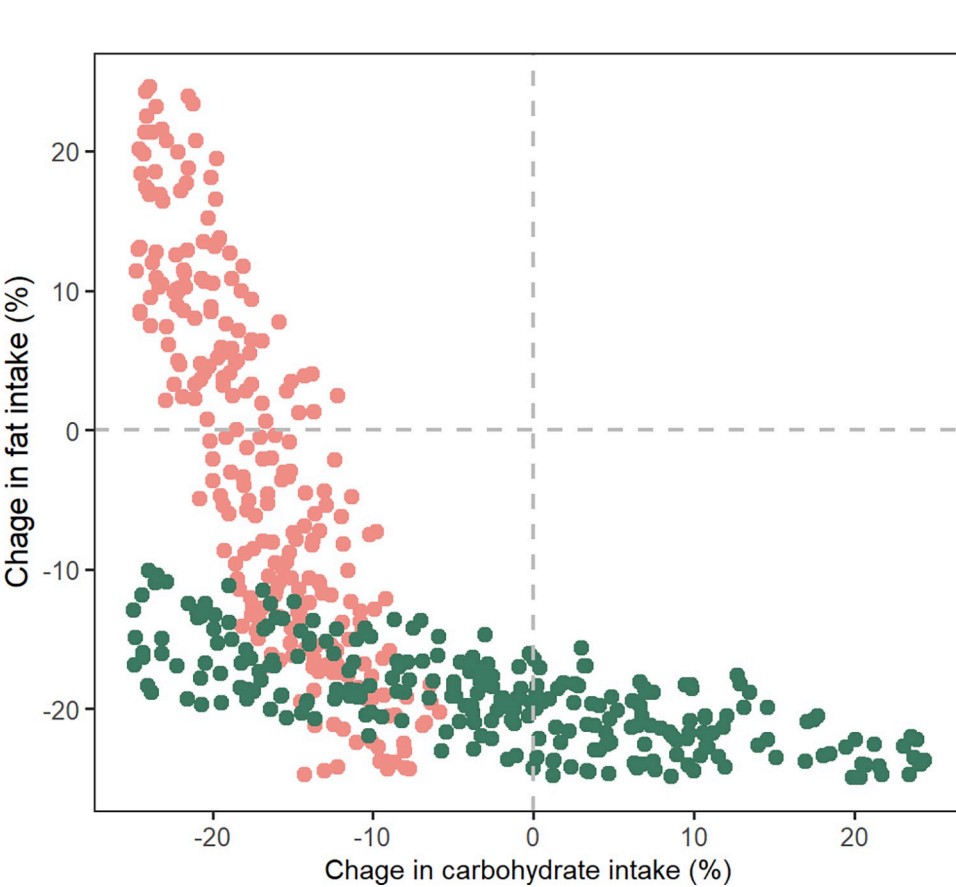

**Fig 5. Optimal changes in carbohydrate and fat intake for targeted weight reduction.** Monte Carlo simulations identified a unique set of "optimal" carbohydrate and fat changes required for each subject that were predicted to lead to a targeted 5–7% reduction in body weight.

## Individuals show differential sensitivity to carbohydrate and fat changes

The simulation-based diet optimization results were used to explore whether all subjects could be classified into carbohydrate or fat sensitive categories. After finding the set of optimal diets for each subject using simulations as described above, lines of best fit were obtained for each subject's (N = 47 subjects with >1 optimal diets) predicted set of optimal diet changes (Fig 7). These lines approximate the predicted optimal diet change patterns for each subject and are a reasonable simplification for easy visualization and analysis of the diet patterns. All lines had negative slopes implying that if a subject were to shift to a smaller reduction in carbohydrate intake, it could be compensated by a larger reduction in fat intake, and vice versa. Additionally, the shifts would have to move along the line, so the magnitude of compensation required was different for each subject as determined by the slope of the line.

For a hypothetical subject whose line of best fit has slope of -1 (angle of -45° with the x-axis), a downward (upward) shift of X% in carbohydrate change could be compensated by a corresponding upward (downward) shift of exactly X% in fat change. Therefore, a subject with a slope of exactly -1 can be considered to be equally sensitive to changes in carbohydrate and fat intake. As the line becomes increasingly horizontal (angle with the x-axis between -45° and 0°, slope between -1 and 0), the sensitivity regime shifts towards greater sensitivity to fat

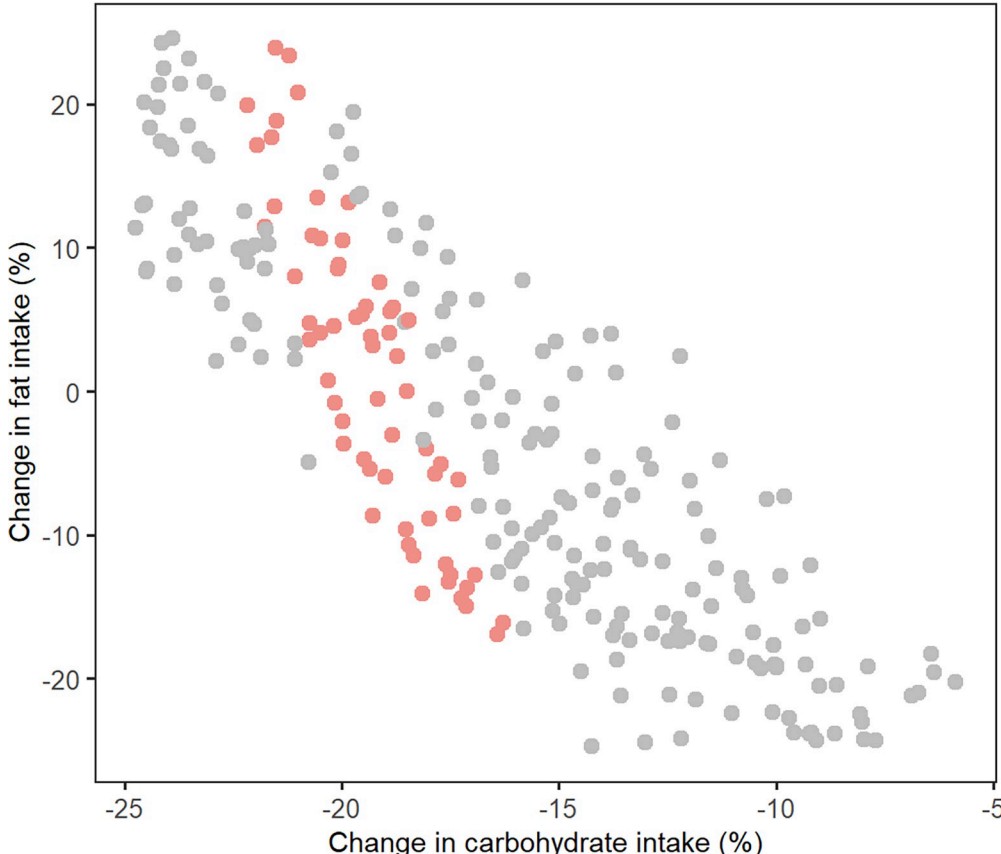

**Fig 6. Including additional biomarker targets further narrows the predicted optimal diets.** The subset (pink circles) of optimal diets identified for subject Test-041 (Fig 5) to achieve a 5–7% reduction in body weight (gray and pink circles) was predicted to additionally reduce HbA1c by 0.1–0.2%.

change because for a nearly horizontal line, fat change must be tightly controlled while carbohydrate change can vary widely. Conversely, as the line becomes more vertical (angle with the x-axis between -90˚ and -45˚, slope < -1), it indicates a greater sensitivity to carbohydrate change. Based on these concepts, individuals were classified as carbohydrate sensitive (slope < -1) or fat sensitive based on the slopes of their lines (slope > -1) (Fig 7). A total of 29 (62%) subjects were identified as has having a greater sensitivity to fat change and 18 (38%) as being more sensitive to carbohydrate changes based on the sensitivity criteria defined above.

## Discussion

Diet therapy can be an effective non-pharmacological method to delay or prevent the onset of T2D; however, diet therapy has not been shown to be consistently effective [4–6, 22]. The lack of effectiveness of diet therapy could be due to personalized dietary requirements [7–9]. Previous studies showed that individuals receiving an identical standardized low-energy diet show variability in their weight trajectories [30]. Metabolic heterogeneity among individuals could be due to genetic and epigenetic factors, microbiome, lifestyle, and environmental exposure [31]. Personalized nutrition is a growing area of focus for both patients and experts.

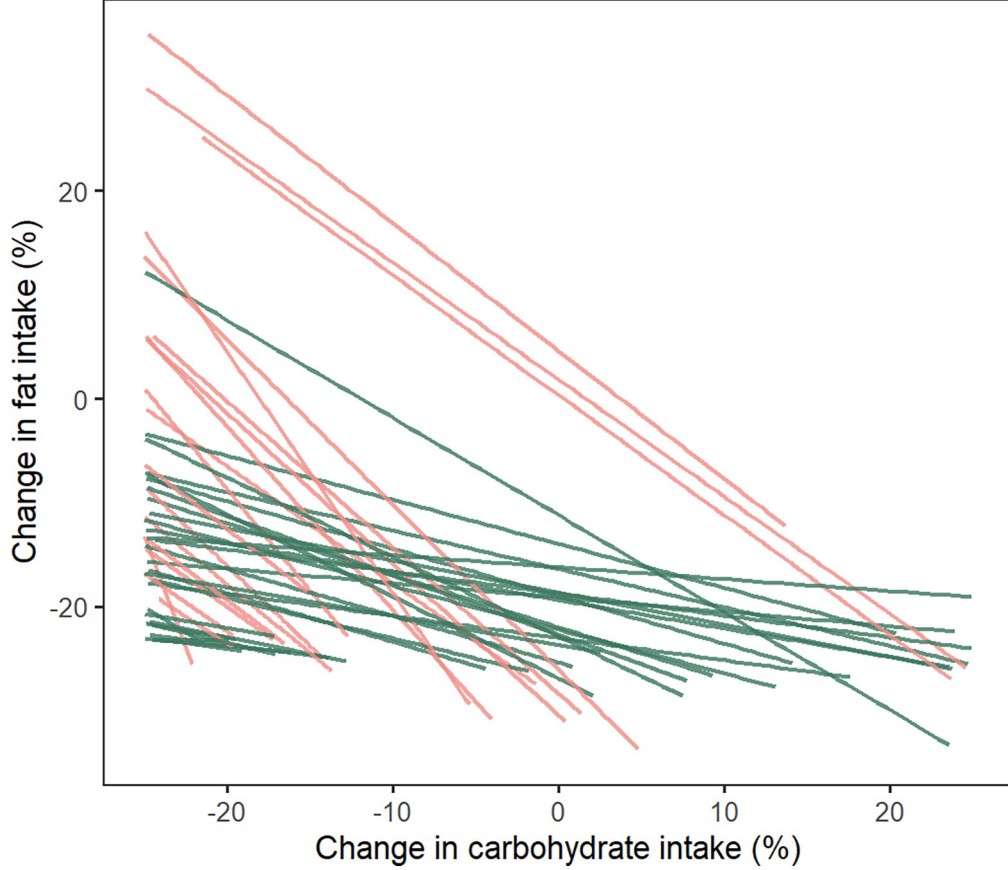

**Fig 7. Optimal diet trajectories and relative sensitivity to macronutrients.** A line was fit to the set of optimal diets predicted for each subject. The slopes of the lines were used to classify subjects into carbohydrate or fat sensitive categories. Lines that tend to be more horizontal (green lines; slope > -1) indicate individuals with greater sensitivity to fat change. Lines that tend to be more vertical (pink lines; slope < -1) indicate individuals with greater sensitivity to carbohydrate change.

Optimizing diet change to individual physiological responses could maximize the impact of lifestyle intervention; however, model-based tools that can automate customization of interventions at the individual level, particularly with a view to achieving long-term health goals, are lacking. We demonstrate, for the first time to our knowledge, the application of a computational simulation model based on physiological mechanisms as a tool to optimize diets for prediabetic individuals with long-term health goals in mind. There are other examples that are less mechanistic and more data-driven, e.g., methods based on machine learning approaches, or those that address shorter-term changes in biomarkers [32–34]. The approach presented in this paper distinguishes itself by using an explainable, mechanistic model and focusing on optimizing diet for long-term goals over multiple months to years.

The emergence of digital twins and digital representation of objects or individuals provides a new opportunity to tailor individualized interventions [35]. We used a previously developed and tested mechanistic simulation model of human physiological processes involved in the onset and progression of diabetes to create digital twins of a subset of pre-diabetic subjects from the J-DOIT1 study. In the default setting, the parameters of the model are calibrated to

represent a "typical" individual. When individual-level time-course data, such as body weight and HbA1c level over time are available, selected parameters of the model can be calibrated to fit the model to an individual subject's data, which leads to a model customized to the subject, i.e., a digital twin of the individual. The digital twin provides a platform to conduct computational experiments quickly and safely in an *in silico* environment [36]. Digital twins were utilized in this study to explore and optimize lifestyle recommendations through simulation.

We leveraged the simulation model to understand the inter-individual variability in responses to lifestyle interventions in the J-DOIT1 study. The selected individuals from the intervention arm were baseline matched with the participants from the control arm of the study. The baseline-matched individuals from the control arm formed the training set (n = 53), and individuals selected from the intervention arm comprised the test dataset (n = 59). Each subject from the training set was calibrated using the simulation model to generate a unique combination of physiological and lifestyle parameters that defined the digital twin of that subject. A key assumption in our approach was that individuals with similar baseline characteristics (age, sex, height, weight, and HbA1c) have similar physiological parameters and historical lifestyle. Therefore, physiological, and historical lifestyle parameters were replicated for the test subjects within each baseline-matched train-test pair. The implication of this assumption is that two individuals who are sufficiently similar at a given point in time are taken to have reached that state through a similar trajectory. Moreover, their future responses to similar lifestyle changes are also implicitly assumed to be similar. In reality, it is possible that two individuals with similar baseline characteristics achieve that state through very dissimilar paths and respond differently to similar interventions. However, this approach is not fundamentally dissimilar from clinical studies that use baseline-matched study arms to compare treatment effects. In the absence of rich historical data about the study subjects, this is a reasonable simplifying assumption to make. From the modeling perspective, this assumption reduced the number of parameters that needed to be estimated for the test subjects and made parameter estimation more tractable. The digital twins generated under this assumption, captured the individual-level dynamics of the body weight with an error of 1.1% (±1.0) and HbA1c levels with an error of 1.4% (±1.4) relative to the actual measurements over a follow-up period of approximately 4 years.

The goodness of fit metrics presented above were calculated across best and worst responders. In the interval between study baseline and first follow-up, which corresponds with the duration of active intervention in the J-DOIT1 study, a wide range of weight dynamics was observed. Some individuals lost nearly 10 kg of body weight while some others gained upward of 5 kg (Fig 4). Other individuals were distributed fairly regularly between these extremes. The goodness of fit metrics indicate that the model captured not only the extreme ends of the response, but also the degrees of response in between. The model captured the dynamics of the period up to the first follow-up, which was generally the period with the largest magnitude of change in body weight and HbA1c for most individuals. Furthermore, the long-term follow-up period of an additional 3 years (approximately) after the first intervention, where the changes were often gradual and less pronounced, were also captured well by the model. Collectively, these observations illustrate that the model performed well at predicting large and small body weight and HbA1c changes over durations of 4 years or more.

The digital twins enabled the exploration of inter-individual variability in response to diet intervention. The digital twins created using the model were used to estimate the actual lifestyle change in terms of total caloric intake for all individuals in the training and testing datasets. We observed that the measured change in body weight generally increased with the model-predicted increase in daily caloric intake; however, similar changes in daily calories were predicted to result in a relatively wide band of weight change (approximately ±2.5 kg

around the line of best fit, or a range of 5 kg) (Fig 4). Considering that the model predicts body weight within a ±1.0 kg range, ±2.5 kg is a relatively wide band of weight change suggesting that changes in calories alone may not be sufficient to precisely predict individual-level changes in body weight. This inter-individual variability in response despite very similar caloric change is reflective of the differences in the underlying physiological parameters of the model for different individuals. As described in greater detail in [20], non-linear interactions between individual physiological parameters result in complex behaviors and lead to variability in response to similar perturbations. This further elucidates the significance of physiology, among other factors, in determining an individual's response to diet.

Just as exogenous lifestyle factors were not fully predictive of the outcome, endogenous physiological parameters were also not found to be correlated with the outcome. This suggests that the outcome of a lifestyle change is an emergent property of complex interactions between underlying physiological processes and exogenous changes. Predicting such a response, therefore, requires an understanding of the complex interactions driving the response. Our physiology-based, quantitative framework, which captures such interactions by design, is well-suited for this purpose.

Having tested the model's ability to satisfactorily describe individual-level dynamics of body weight and HbA1c, we applied the model to generate optimal diet recommendations for individuals in the training and testing datasets. Monte Carlo simulations were performed for individuals using their digital twin, and a unique set of "optimal" carbohydrate and fat changes required for a targeted 5–7% reduction in body weight was determined. The model predicted that a range of diets unique to each individual could help achieve this goal, and there is no single ideal diet to achieve the target body weight. Analysis of optimal diet trajectories at the subject levels suggested that while some patients required tight control over fat intake (individuals sensitive to fat change), others required a greater focus on managing carbohydrate intake (individuals more sensitive to carbohydrate change). A few individuals in Fig 7, those that are in the bottom left corner, appear to have a narrow range of optimal diets. However, that is an artifact of the sampling space (-25% to 25% of baseline value) available for carbohydrate and fat intake. The lines are cutoff at these boundaries because greater changes were not explored along either axis. Individuals in the bottom left corner appear to require rather large reductions in both carbohydrates and fats to achieve the target weight loss, and only a small number of diets in the sampled space fit that description. The set of optimal diets found to meet the weight reduction goal could be further refined by including additional goals, e.g., a targeted reduction of 0.1–0.2% points in HbA1c. Even though the range chosen for targeted reduction in HbA1c is close to the expected measurement error for HbA1c, any threshold could have been chosen for this computational experiment without qualitatively altering the conclusion. These results support the role of personalized nutrition and dietary recommendations in improving health outcomes and demonstrate the potential utility of our approach in identifying such personalized recommendations based on historical subject data.

The modeling and analyses presented in this work are affected by a few limitations of data and methodology that should be acknowledged. The target population of our analysis only included Japanese individuals with prediabetes, thus limiting the generalizability of the predictions. The matching algorithm used to create pairs of train-test subjects allowed a small degree of mismatch so that matched pairs could be practically found. The assumption of physiological identity between the matched pairs has, therefore, some inaccuracy inherent to it and could impact the estimation of parameters as well as model predictions. Furthermore, all lifestyle changes were simulated as step functions, as this was mathematically the simplest form in the absence of additional information on individual lifestyle habits. In real life, lifestyle factors may be much more variable and may follow trends very different from a step function. This

assumption is likely to impact the timing and rate of change of model-predicted variables like body weight. Another concern with mechanistic models is that of parameter identifiability [37]. With our piecewise calibration approach, there can be up to 4 lifestyle changes throughout the observation period. This leads to a total of up to 12 parameters (4 changes x 3 parameters/change) that can be adjusted in the worst-case scenario. On the flip side, there are up to 12 observations per subject (6 each for body weight and HbA1c) for more than 75% of subjects. This provides some assurance that, typically, the number of estimated parameters and the number of data points are relatively balanced. However, this does not completely defend against practical identifiability issues as highlighted by Raue et al. Finally, the mechanistic mathematical model used in this study makes several assumptions about the physiological processes underlying diabetes onset and progression, which may not always reflect the underlying biology and physiology accurately. Nevertheless, even with these limitations, the model predicted the body weight and HbA1c time courses of the training as well as test groups with high accuracy, which lends credence to the model and supports its use for predictive analysis.

An advantage of the model-based framework developed in this study over approaches like precision nutrition is that it can provide optimal dietary recommendations without requiring specific genetic and microbiome data, making it a quicker, lower-cost alternative. Prior validation of the simulation model using long-term data [20] and additional validation in this work using a subset of participants from the J-DOIT1 study showed that the model predicts weight changes and glycemic control in individuals with high accuracy. This provides assurance that the framework can be used to predict optimal dietary recommendations for prediabetic individuals. Further validation through additional retrospective analyses and prospective studies in human subjects is required to increase confidence in this simulation modeling framework and confirm its utility in clinical practice.

Validation could be carried out in a few different ways, e.g., by testing metabolic fluxes in the model or by testing predictions of clinical endpoints such as body weight. Testing fluxes is extremely challenging because thorough measurement of fluxes through the major molecular species included in the model in patients is not trivial. A more realistic and practical approach is to test model predictions of clinically relevant outcomes (e.g., body weight, HbA1c, glucose, etc.) for different subject cohorts (e.g., variability in race/ethnicity, sex, age, metabolic health, etc.). This would allow easier testing of conveniently measurable outcomes that are clinically relevant and familiar for both clinical practitioners and their patients alike.

In the process of training and testing models such as the one presented here, historical patient data (e.g., results of regular physical examination), real-world information about lifestyle factors (e.g., physical activity recorded through digital wearable devices), and self-reported information (e.g., digital records of dietary habits in a phone application) could be very valuable in further constraining the model fits and improving long-term predictions. Obtaining, harmonizing, and cleaning this kind of data, however, is a non-trivial problem. Furthermore, the reliability of self-reported data has been challenged and could end up complicating the analysis instead of helping it [38]. Objective clinical measurements, such as blood biomarkers and body weight, do not suffer from these limitations. Rich information about such objective measurements could significantly improve the predictive ability of the model.

The latest Dietary Guidelines for Americans (DGA) focus on limiting fat, especially saturated fat, and allowing higher carbohydrate intake. Volek *et al*. have argued that the DGA recommendations of a low-fat high-carbohydrate diet for the past several years have coincided with rapidly escalating epidemics of obesity and T2D that contribute to the progression of cardiovascular diseases [39]. This guideline lacks flexibility and does not appreciate the heterogeneity in individuals' responses to dietary interventions. The findings of the J-DOIT1 study, coupled with the model-based framework for diet optimization presented in this study, offer

additional evidence to convince experts and policymakers of the need for individually optimized diet interventions because of inter-individual variability in responses to identical diets. Our modeling framework can simulate changes in body weight and glycemic control as a result of lifestyle interventions at an individual level. The ability to optimize nutritional strategies using this model could help dieticians and physicians personalize diet recommendations to their patients' goals.

## Supporting information

**S1 Fig. Calibrated pairs for best responders, low call frequency group.**
(TIF)

**S2 Fig. Calibrated pairs for best responders, medium call frequency group.**
(TIF)

**S3 Fig. Calibrated pairs for best responders, high call frequency group.**
(TIF)

**S4 Fig. Calibrated pairs for worst responders, low call frequency group.**
(TIF)

**S5 Fig. Calibrated pairs for worst responders, medium call frequency group.**
(TIF)

**S6 Fig. Calibrated pairs for worst responders, high call frequency group.**
(TIF)

**S7 Fig. Goodness of fit assessment.** Model-predicted body weight and HbA1c values for all subjects across time points show reasonable concordance with corresponding measured values with most values lying on or close to the line of identity.
(TIF)

**S8 Fig. Model-estimated change in calories vs. measured change in HbA1c from baseline.** The measured change in HbA1c from baseline to the first follow-up during the J-DOIT1 intervention plotted against model-estimated change in calories per day due to both diet and exercise changes averaged over the same period for subjects in the intervention arm. The gray number in each quadrant is the fraction of data points in that quadrant. The data points fit a linear regression model (solid gray line) with $r^2 = 0.20$ and a residual standard error of 0.28 points.
(TIF)

**S9 Fig. Distribution of randomly sampled total calorie change.** Total caloric changes sampled to find optimal diets are shown for three randomly selected subjects (Test-020, Test-022, and Test 026). Total caloric change is approximately normally distributed with mean 0 and covers the ±25% range.
(TIF)

**S10 Fig. Relationship between macronutrient change and weight change.** All sampled macronutrient changes (N = 2000 per subject) were plotted against the model predicted weight change for three randomly selected subjects (same subjects as in S9 Fig). Carbohydrate and fat changes are, on average, monotonically related with weight change; however, protein changes are not correlated with weight change.
(TIF)

**S1 File.**
(ZIP)

## Author Contributions

**Conceptualization:** Momoko Fukasawa, Paul D'Alessandro, Gaurav Dwivedi.

**Data curation:** Julia H. Chen, Gaurav Dwivedi.

**Formal analysis:** Julia H. Chen, Momoko Fukasawa, Shikhar Pandey, Sai Phanindra Venkatapurapu, Gaurav Dwivedi.

**Investigation:** Naoki Sakane.

**Methodology:** Sai Phanindra Venkatapurapu, Gaurav Dwivedi.

**Project administration:** Naoki Sakane, Hideshi Kuzuya.

**Supervision:** Naoki Sakane, Hideshi Kuzuya, Paul D'Alessandro, Gaurav Dwivedi.

**Validation:** Julia H. Chen, Momoko Fukasawa, Shikhar Pandey, Paul D'Alessandro, Sai Phanindra Venkatapurapu, Gaurav Dwivedi.

**Visualization:** Julia H. Chen, Momoko Fukasawa, Sai Phanindra Venkatapurapu, Gaurav Dwivedi.

**Writing – original draft:** Naoki Sakane, Sai Phanindra Venkatapurapu, Gaurav Dwivedi.

**Writing – review & editing:** Julia H. Chen, Momoko Fukasawa, Akiko Suganuma, Paul D'Alessandro, Sai Phanindra Venkatapurapu, Gaurav Dwivedi.

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
