## [Decision Letter · Decision Letter 0]

13 Jul 2023

PONE-D-23-14749Optimization of nutritional strategies using a mechanistic computational model in prediabetes: Application to the J-DOIT1 study dataPLOS ONE

Dear Dr. Sakane,

Thank you for submitting your manuscript to PLOS ONE. After careful consideration, we feel that it has merit but does not fully meet PLOS ONE’s publication criteria as it currently stands. Therefore, we invite you to submit a revised version of the manuscript that addresses the points raised during the review process.

We look forward to receiving your revised manuscript.

Kind regards,

Jianhong Zhou

Staff Editor

PLOS ONE

Journal Requirements:

"This work was supported by JSPS KAKENHI Grant Number 18k01988.The funders had no role in study design, data collection and analysis, decision to publish, or preparation of the manuscript.

PricewaterhouseCoopers, LLP provided support in the form of salaries for the following authors - [JHC, MF, SP, PMD, SPV, GD] but did not have any additional role in the study design, data collection and analysis, decision to publish, or preparation of the manuscript."

Reviewers' comments:

Reviewer's Responses to Questions

**Comments to the Author**

1. Is the manuscript technically sound, and do the data support the conclusions?

Reviewer #1: Partly

Reviewer #2: Yes

2. Has the statistical analysis been performed appropriately and rigorously? 

Reviewer #1: Yes

Reviewer #2: N/A

3. Have the authors made all data underlying the findings in their manuscript fully available?

Reviewer #1: Yes

Reviewer #2: No

4. Is the manuscript presented in an intelligible fashion and written in standard English?

Reviewer #1: Yes

Reviewer #2: Yes

5. Review Comments to the Author

Reviewer #1: Summary:

The manuscript proposes a novel, mechanistic model-based analysis of the inter-individual variability in long-term dietary responses. In addition, it presents the use of the estimated personalized models (“digital twins”) to hypothesize optimal diet composition with respect to carbohydrate and fat percentage. The manuscript deals with the challenging methodology related to the lifestyle-based prevention and treatment of diabetes which is an increasingly important societal issue. In this way, the manuscript fits into the highly relevant discussion about personalized/precision nutrition and digital twinning.

In general, the study is an innovative take on model-based personalized/precision nutrition and may very well be in the interest of the broader readership of PLOS ONE. However, there are some methodological choices and details that are unclear in the current state of the manuscript. In addition, the discussion section would benefit from an update to make sure not to include redundancies (particularly with the results section).

I have included specific commentaries below:

Major remarks:

1. It is unclear how the parameter estimation of the lifestyle parameters is carried out. Is the model simulation stopped at the discrete moments of the lifestyle changes, the initial conditions set to the end of the previous segment and then the parameters are estimated to fit the segment until the next lifestyle change? How is the timing of the lifestyle changes determined? Is it based on the measurement times for the individuals? These details are necessary for the reproducibility of the results. (related to p14 “Calibration of the test dataset”)

2. Following from the previous question, how much data is used in calibrating the parameters? I.e. if the procedure is as hypothesized in q1, is there enough data to uniquely identify the value of the parameters? As far as I understand, there is a single measurement of HbA1c and weight for a segment (T), and yet 3 parameters (CI, FI, ∆PA) are estimated for each segment? In general, analysis of and remarks about parameter identifiability would benefit the manuscript. Practical unidentifiability may lead to not well determined model simulations (e.g. see Raue et. al, 2009, https://doi.org/10.1093/bioinformatics/btp358).

The data section of the methods should be updated with the exact measurements (including their quantity in terms of time points etc.) used to estimate the model (related to p14 “Calibration of the test dataset”)

Continuing from this, throughout the manuscript the authors suggest that their model may be used to help dieticians and physicians to personalize diet recommendations. However, I believe that in order to make this a reality, a thorough validation of the model fluxes would be necessary. For example, when forcing the variability into the 3 estimated lifestyle parameters, do the fluxes still represent physiologically valid values?

3. Parameters were estimated for carb intake, fat intake, protein intake, and physical activity in the analysis presented in the manuscript. However, the only results presented were about the total caloric change and its effect on predicted weight, and the fat and carb intake changes to achieve an optimal diet. What about proteins and physical activity? If those parameters did not influence the outcomes, then why estimate them? If they did, what do the results look like?

Minor remarks:

4. Fig 3 may be an optimistic representation of the model fits after looking at all of the ones in the supplement. In general, I accept the model performance (the trends seem to be captured for the majority of the fits), it is perhaps more representative to give a few examples in the main text instead of the single good fit that is shown in Fig 3. The results at lines 286-288 should also be updated to reflect the above.

5. P17/line 329: “This suggests that changes in calorie intake alone may not be sufficient to predict individual-level changes in body weight.” Do the authors mean the total caloric change (including the change in physical activity)?

6. P16/line 315: Why change to first follow up only? I assume because of the largest observed effect?

Following from this, perhaps some discussion could be added on how to deal with the individuals who are not in the extreme responders of the data set (i.e. the remaining ~2500 individuals of the J-DOIT1 data set)? I believe the societal challenge is to find improvements for these individuals. Could it be hypothesized that given more rich data (additional/different measurements e.g. detailed food diaries; blood biomarkers etc.) could enable the model to be used for individuals with less pronounced responses? The selected 112 individuals serve nicely as proof-of-concept, but is it even a realistic representation of the J-DOIT1 data, not to mention the Japanese prediabetic population?

7. P17/line 323: The figure references on P17/line 323 is unclear, is this referring to S7? The figure labels indicate measured and predicted body weight and not caloric increase.

8. P17/line 326-331: “This suggests that changes in caloric intake alone may not be sufficient to predict individual-level changes in body weight”. This statement assumes no uncertainty in the model predicted calorie change. In order to truly assess this, it would be good to know the prediction uncertainty. In addition, this statement is better suited to the discussion (where it’s also repeated, therefore I would remove it from the results).

9. P18/line 344-345: no physiological parameters were associated with changes in body weight or HbA1c. Did the authors try Spearman’s correlation to test this or only Pearson? This lack of correlation is not surprising from the experimental design (only lifestyle parameters are allowed to change). However, from a biological perspective, I would expect that parameters related to things such as insulin sensitivity to improve in successful interventions. Perhaps it’s worth discussing this limitation of the study design in the discussion.

10. P18/line 359: “no single optimal diet..”; do the authors mean that the optimal diet is not unique but different diets can achieve the same effect? It is stated, that 11 subjects did not see an improvement in the expected range (5-7% reduction in body weight). Can this be because of chance, where the sampled diet changes complemented the total caloric balance (e.g. small reduction in fat and carb but large increase in protein)?

Regarding the experiment about analyzing hypothetical optimal diets (“Diet therapy is predicted to have maximal effectiveness when optimized individually”): More details about the experimental setup should be clarified and included in the Methods section. How many random simulations were carried out per individual? How was the MCMC set up? Why not simple latin-hypercube sampling (since there’s only 3 parameters being changed)?

11. P19/line 379-380: the 0.1-0.2% reduction in HbA1c is very close to the expected measurement error. In practice, this does not change the modelling result, but it should be mentioned.

12. In addition to the results already discussed on Fig 5. It is interesting to see the variation in “flexibility”. Some individuals seem to have a very narrow range of optimal diets. This also seems to correspond with how sensitive they are to carbs or fats, i.e. lines closer to a slope of -1 are longer (?).

13. P21/line 428: “however, tools that can enable customization of interventions at the individual level are lacking.” This may be very controversial since dietician advice can already be consider as customized to the individual. Maybe rephrase as “automated” or “model-based” customization or similar.

14. P21/line 429-430: This is a very strong statement. See https://doi.org/10.21203/rs.2.20798/v1;
https://doi.org/10.1016/j.cell.2015.11.001; Of course, here the temporal scale is long term as opposed to short term (meal responses) and indeed the model may be argued to be mechanistic instead of fully data-driven. However, these distinctions should be made more explicit, as the general topic of using computational models to optimize diet is not new.

15. P22/lines 444-458: instead of reiterating the experimental setup from them methods, perhaps include some discussion on how the choice of matching the training and test subjects may affect the interpretation of the results.

16. P23/line 470: It is stated that exogenous lifestyle factors were not fully predictive of the outcome. It is unclear what is meant by this. As far as I understand, the exogenous (lifestyle facotrs) parameters were the only thing changed in order to fit the response. Therefore, they must be predictive of the outcome, are they not? In addition, in this paragraph it is mentioned that the presented framework allows to capture the various interactions therefore it is well suited to understand the interactions. However, the study of the interactions is severely limited by the choice of fixed and varied parameters. From a biological perspective, on the scope of 4 years insulin resistance should also change in combination with some of the lifestyle parameters. This should be addressed. With more data, would it be possible to also estimate these parameters and truly uncover the interactions? What data would be ideal and in what quantity?

17. P25/line 517-520: these are more limitations (some are reiterated), move them to the appropriate paragraph about limitations.

18. Did the authors consider sharing an implementation of their model? Making it available through github (or equivalent) could increase the reach and impact of the authors’ work.

19. Finally, please do a careful check for typos etc. The figure references seem to be not working and there are also some tracked changes left in the manuscript. Particularly, in the discussion there were many inconsistencies.

Reviewer #2: There are one or two small errors in the text but none introduce ambiguity. eg Figure 5 x axis is misspelt. Please check the whole paper thoroughly. Otherwise it was a very well written paper.

This is a very well written paper. I found it difficult to read though because it is so information dense and the modelling language used is not very familiar to me, so it is my limitation as a referee of this type of paper that is the problem, and nothing wrong with the paper that I could see. I'm sure it will be understandable to a reader familiar with modelling in the health area. Because it is so well written, and the results are clearly presented I would be in favour of letting competent readers judge for themselves.

It seems that familiarity with previous J_DOIT1 would be required to fiully understand the paper.

The discussion is concise and the limitations of the study are well presented. The referencing seems to be thorough.

I responded "no" to question 3 because I though there should be a statement of data availability at the end of the paper.

6. PLOS authors have the option to publish the peer review history of their article (what does this mean?). If published, this will include your full peer review and any attached files.

Reviewer #1: **Yes: **Balazs Erdos

Reviewer #2: No

---

## [Author Response · Author response to Decision Letter 0]

13 Sep 2023

RESPONSES TO REVIEWERS’ COMMENTS

We thank the reviewers for their thoughtful comments. We greatly appreciate the time and effort the reviewers have spent in reviewing this manuscript. Our responses to the comments are provided below. We have also updated the manuscript in several places to address the reviewers’ comments, and the key changes are highlighted in our responses below.

Please note that the page and line numbers given in the responses are for the tracked version of the manuscript with all markups shown (using the “All Markup” option selected in the Review tab of Microsoft Word).

We wrote our responses using Microsoft Word, where we denote the authors’ responses using red text and have included some figures. The colored text and figures do not carry over to the plain text version of the responses required by the submission system. The version with figures and colored text appears towards the end in the PDF rendered by the submission system. The reviewers might find it more convenient to read that version instead of the plain text version.

Reviewer 1

The manuscript proposes a novel, mechanistic model-based analysis of the inter-individual variability in long-term dietary responses. In addition, it presents the use of the estimated personalized models (“digital twins”) to hypothesize optimal diet composition with respect to carbohydrate and fat percentage. The manuscript deals with the challenging methodology related to the lifestyle-based prevention and treatment of diabetes which is an increasingly important societal issue. In this way, the manuscript fits into the highly relevant discussion about personalized/precision nutrition and digital twinning.

In general, the study is an innovative take on model-based personalized/precision nutrition and may very well be in the interest of the broader readership of PLOS ONE. However, there are some methodological choices and details that are unclear in the current state of the manuscript. In addition, the discussion section would benefit from an update to make sure not to include redundancies (particularly with the results section).

First, the authors thank Dr. Erdős for his very detailed and thoughtful review of the manuscript. It is evident that he has applied his expertise of the field to critically review our work, and the authors are of the opinion that incorporating his feedback will significantly improve the clarity and quality of our manuscript. We thank the reviewer for acknowledging the societal relevance and scientific value of the topic and methodology presented in our manuscript. Specific major and minor comments are addressed pointwise below. 

--

I have included specific commentaries below:

Major remarks:

1. It is unclear how the parameter estimation of the lifestyle parameters is carried out. Is the model simulation stopped at the discrete moments of the lifestyle changes, the initial conditions set to the end of the previous segment and then the parameters are estimated to fit the segment until the next lifestyle change? How is the timing of the lifestyle changes determined? Is it based on the measurement times for the individuals? These details are necessary for the reproducibility of the results. (related to p14 “Calibration of the test dataset”)

The reviewer’s point is well taken. There is some text in the section titled “Calibration of the test dataset” that broadly explains the methodology used, but we acknowledge that it is not detailed enough. The reviewer’s conjecture is partly correct – the simulation is stopped at discrete time points, lifestyle parameters are changed, and the simulation continues on using the final state of the last segment as the initial state of the new segment. The timing of when the simulation start-stop occurs was, as explained in the original text, determined empirically by manually observing the trends in body weight and HbA1c. If, for instance, both weight and HbA1c showed a decreasing trend over multiple observations, it was assumed that a single diet change at the beginning of the observation period should be able to explain all the observations. If it was determined after calibration that the observations could not be satisfactorily explained with a single diet change, then the segment was broken into two parts and an additional diet change was introduced. Up to 4 lifestyle changes were allowed per subject over the entire observation period of nearly 4 years. In the worst-case scenario, this could theoretically lead to one diet change per observation because a small fraction of subjects had only 4 observations over the entire follow-up period for one of body weight or HbA1c. This, however, did not occur in practice. More than 75% of the subjects had 6 observations for either one or both of body weight and HbA1c; nearly 90% of the subjects had 5 or more observations. 7 subjects required 4 discrete lifestyle changes to explain all their data, but all 7 subjects had 6 measurements for body weight and HbA1c (except for 1 subject who had 5 HbA1c measurements). Therefore, a scenario in which there is one diet change per observation did not occur. As mentioned, this process was manual (did not involve any automation) and a conscious effort was made to minimize the number of lifestyle changes required to explain the entire observation period. We acknowledge that this does not guarantee that the number of lifestyle changes introduced was indeed the minimum possible number for each subject. More details around numbers of time points and lifestyle changes have been added to manuscript in the Model Calibration and Testing section.

When it comes to fitting the lifestyle parameters, each segment was not fit individually. Instead, the entire time course from baseline to the end of the 1-year follow-up period was fit collectively and all lifestyle parameters at the pre-determined change points were allowed to vary collectively by the fitting algorithm. This approach was chosen keeping in mind that history of lifestyle changes can have a bearing on the future response to additional lifestyle changes. Concretely, if the lifestyle change in one segment is fixed at a particular value, then the number of lifestyle parameters that can explain the next segment is also limited by the choice of that value. Instead of constraining the optimization in this way, we chose to allow all lifestyle parameters to float collectively during. Our rationale for choosing this method over the one that fits each segment before moving on to the next one was two-fold:

1) This method gives the system more flexibility to capture the general trends over the entire observation period, even if the fitting is not very precise. It is possible that in the approach where each segment is fitted separately before moving on to the next one, we could end up with a lifestyle choice in segment 1 that makes it very hard to explain the changes in segment 2 with a reasonable lifestyle change.

2) Fitting each segment individually increases the risk of overfitting (as the reviewer points out in the next comment). The number of free parameters in such a situation would, at least for some segments, far exceed the number of data points.

It is possible that in practice the two approaches (fitting one segment at a time versus fitting all lifestyle parameters collectively for the entire time course) yield similar results. Nonetheless, this is a design choice we made based on the reasons presented above.

Text has been added to the methods section to further explain the process (pg10, lines 231-241; pg13, lines 271-272 and 274-281).

--

2. Following from the previous question, how much data is used in calibrating the parameters? I.e. if the procedure is as hypothesized in q1, is there enough data to uniquely identify the value of the parameters? As far as I understand, there is a single measurement of HbA1c and weight for a segment (T), and yet 3 parameters (CI, FI, ∆PA) are estimated for each segment? In general, analysis of and remarks about parameter identifiability would benefit the manuscript. Practical unidentifiability may lead to not well determined model simulations (e.g. see Raue et. al, 2009, https://doi.org/10.1093/bioinformatics/btp358).

We acknowledge and appreciate the reviewer’s concern about parameter identifiability issues that could affect our analysis. As the reviewer would appreciate and as pointed out in the reference cited by the reviewer, this is a common issue with mechanistic/system models. Our response above to the previous comment provides an explanation of how we attempted to limit the problem of having too many parameters with too few observed data points. Of course, this does not fully eliminate the unidentifiability problem. With our approach there can be up to 4 lifestyle changes throughout the observation period. This leads to a total of 12 parameters (4 changes x 3 parameters/change) that can be adjusted in the worst-case scenario. On the flip side, there are up to 12 observations per subject (6 each for body weight and HbA1c) for more than 75% of subjects. This provides some assurance that, typically, the number of estimated parameters and the number of data points are relatively balanced. Nonetheless, this does not completely defend against practical identifiability issues as highlighted by Raue et al. 

As suggested by the reviewer, we have added text in the Discussion section to acknowledge potential issues with identifiability (pg28, lines 676-683).

--

2.1 The data section of the methods should be updated with the exact measurements (including their quantity in terms of time points etc.) used to estimate the model (related to p14 “Calibration of the test dataset”)

Our interpretation of this comment is that the reviewer would like to see statistics about the number of data points used to estimate the parameters. We have added information about the number of body weight and HbA1c measurements available per subject and the number of parameters estimated per subject in the “Calibration of the test dataset” section (pg13, lines 274-281). 

--

2.2 Continuing from this, throughout the manuscript the authors suggest that their model may be used to help dieticians and physicians to personalize diet recommendations. However, I believe that in order to make this a reality, a thorough validation of the model fluxes would be necessary. For example, when forcing the variability into the 3 estimated lifestyle parameters, do the fluxes still represent physiologically valid values?

We agree with the reviewer, and as acknowledged in the Discussion (line 518 of the original submission; ), that further clinical validation of the model, ideally prospective, is essential before it can be used for practical applications. However, we are of the opinion that validation of metabolic fluxes, as suggested by the reviewer, is extremely challenging because thorough measurement of fluxes through all the molecular species included in the model is not practically feasible. A more realistic approach is to test model predictions of clinically relevant outcomes (e.g., body weight, HbA1c, glucose, etc.) for different subject cohorts (e.g., racial/ethnic variability, sex, age, metabolic health, etc.). This approach has the advantage of being more practical in the clinical setting as it relies on easily measurable and outcomes that can be conveniently tested against model predictions. Moreover, endpoints like body weight and HbA1c are clinically relevant and familiar for both clinical practitioners and their patients. Simultaneously testing multiple clinical endpoints will also provide assurance that the internal dynamics of the model, including interactions and fluxes, result in dynamics that appropriately mimic reality.

We have previously tested long-term predictions of the model using data from a US patient cohort (ref to initial paper). Further testing against diverse populations such as that done in this manuscript with a Japanese prediabetic patient population builds confidence in the model’s ability to predict out of sample. We, however, agree with the reviewer that continued testing against additional patient populations, clinical endpoints in retrospective and prospective settings are necessary to further improve and validate the model before it can be used as an independent tool in the clinical setting.

We have added these points in the discussion (pg29, lines 702-709).

--

3. Parameters were estimated for carb intake, fat intake, protein intake, and physical activity in the analysis presented in the manuscript. However, the only results presented were about the total caloric change and its effect on predicted weight, and the fat and carb intake changes to achieve an optimal diet. What about proteins and physical activity? If those parameters did not influence the outcomes, then why estimate them? If they did, what do the results look like?

We would like to clarify that protein intake was not changed during the model calibration process for either the training or the test dataset. The effect of changes in protein intake on body composition are complex, but it is generally accepted that any weight loss induced by increased protein intake is not due to the increased protein intake itself. Rather, it is largely driven by the greater feeling of satiety caused by high protein diets, which leads to overall decreased caloric intake. Under isocaloric conditions, a high protein diet does not lead to a significant change in body weight (see the review by Westerterp-Plantenga and Lejeune; link below this response), a behavior that is captured by the model. Because of this generally accepted knowledge about the role of proteins in body weight regulation, we decided to limit our parameter search to only carbohydrates and fats during the model calibration process. 

When simulating random diet combinations to find optimal diets, however, we did include protein in the simulations because in the simulations we were not dealing with problems of identifiability and there was no strict need to limit the parameter space we were exploring. Furthermore, allowing all three macronutrients to change more closely mimics real life. Physical activity change was not included in these simulations as our focus was diet optimization. Even though protein was included in the simulations, as expected, changes in protein did not show any correlation with body weight changes. We have added Supplementary Figure S10 to illustrate this point and are including it below for easy reference. For 3 randomly selected subjects, the figure shows the scatter of change in macronutrient value against change in body weight. Each subject has a unique, and on average monotonic, relationship of carbohydrate/fat change with body weight change. However, protein change did not show a correlation with weight change for any of the subjects. As illustrated by these examples, the model is structurally designed to limit the effect of protein changes on body weight. Therefore, even though protein was sampled in the diet optimization experiments, it does not confound the interpretation of results with respect to carbohydrates and fats. We have added text in to explain this point (pg20, lines 448-451).

 

Figure. Relationship between macronutrient change and body weight change shown for three randomly selected subjects. Fat and carbohydrate changes were, on average, monotonically correlated with weight change and each subject had a unique structure of this correlation. Protein changes were not correlated with weight change for any subject.

Reference: Protein intake and body-weigh regulation 

https://www.sciencedirect.com/science/article/pii/S0195666305000498)

--

Minor remarks:

4. Fig 3 may be an optimistic representation of the model fits after looking at all of the ones in the supplement. In general, I accept the model performance (the trends seem to be captured for the majority of the fits), it is perhaps more representative to give a few examples in the main text instead of the single good fit that is shown in Fig 3. The results at lines 286-288 should also be updated to reflect the above.

We accept that the example presented in Fig. 3 presents a selected case. As suggested by the reviewer, we have updated the text to represent this fact and referred the reader to specific examples in the supplementary figures for a more comprehensive idea of the quality of fits (pg15, lines 337-344).

--

5. P17/line 329: “This suggests that changes in calorie intake alone may not be sufficient to predict individual-level changes in body weight.” Do the authors mean the total caloric change (including the change in physical activity)?

Yes, this should have been total caloric change. This sentence is now deleted in response to comment #8.

--

6. P16/line 315: Why change to first follow up only? I assume because of the largest observed effect?

Following from this, perhaps some discussion could be added on how to deal with the individuals who are not in the extreme responders of the data set (i.e. the remaining ~2500 individuals of the J-DOIT1 data set)? I believe the societal challenge is to find improvements for these individuals. Could it be hypothesized that given more rich data (additional/different measurements e.g. detailed food diaries; blood biomarkers etc.) could enable the model to be used for individuals with less pronounced responses? The selected 112 individuals serve nicely as proof-of-concept, but is it even a realistic representation of the J-DOIT1 data, not to mention the Japanese prediabetic population?

The reason for choosing the period up to the first follow up was because that is when the J-DOIT1 intervention ended, and the classification of best and worst responders was based on body weight and HbA1c response up to that time point.

We would like to point out that our effort was not limited to explaining the body weight and HbA1c trends over just the first year. As presented in Figure 3 and explained in the Methods section, the model was used to predict body weight and HbA1c changes over the entire follow-up period of the study, which was approximately 4 years for most subjects. It is correct that the largest change in weight and HbA1c was observed in the first year (up to the first follow up) because that period corresponded with the JDOIT intervention; however, please note that the model also explained trends in both biomarkers in the post-intervention, long-term follow up period of nearly 3 years. Notably, in these 3 years, responses were highly variable at the individual level – some subjects remained nearly steady at the post-intervention level, some showed further improvement, while still others showed reversal of improvement. Additionally, the magnitude of these changes in the post-intervention period was also variable. The results demonstrate that the model performed reasonably at capturing all this variability in response over time.

Another way to think about this is that while we selected individuals as best or worst responders based on their outcomes at the first follow-up (end of the JDOIT intervention), the subjects did not necessarily maintain that status throughout the long-term follow-up period. Therefore, not only did the model capture the extremes of the responses in the first year, but also explained the maintenance or departure from those trends in the subsequent three years. We believe that this significantly allays any concerns about the ability of the model to explain responses that do not lie in the extremes. Additionally, it can also be argued that if the model can satisfactorily capture the trends of both extremes, then capturing individuals in between should be feasible for the model, which is demonstrated by the results. 

We agree with the reviewer’s comment that additional data could help to further constrain the model. However, the nature, quality, and quantity of data introduce a lot of subjectivity in this regard. For instance, the validity of self-reported dietary intake remains controversial and could, in some situations, do more harm than good. Having said that, objective measurements including additional blood biomarkers could improve predictive ability for all individuals. We have added this point and key elements of the arguments presented above to the Discussion section (pg25, lines 586-598; pg30, lines 719-728).

--

7. P17/line 323: The figure references on P17/line 323 is unclear, is this referring to S7? The figure labels indicate measured and predicted body weight and not caloric increase.

It is a reference to Figure 4 in the main text. It appears cross reference links were inadvertently broken in many places during the submission process. We have reviewed and updated all references.

--

8. P17/line 326-331: “This suggests that changes in caloric intake alone may not be sufficient to predict individual-level changes in body weight”. This statement assumes no uncertainty in the model predicted calorie change. In order to truly assess this, it would be good to know the prediction uncertainty. In addition, this statement is better suited to the discussion (where it’s also repeated, therefore I would remove it from the results).

We have moved the sentence to discussion and qualified it in the context of average prediction error (pg 26, lines 610-612).

--

9. P18/line 344-345: no physiological parameters were associated with changes in body weight or HbA1c. Did the authors try Spearman’s correlation to test this or only Pearson? This lack of correlation is not surprising from the experimental design (only lifestyle parameters are allowed to change). However, from a biological perspective, I would expect that parameters related to things such as insulin sensitivity to improve in successful interventions. Perhaps it’s worth discussing this limitation of the study design in the discussion.

Reading the reviewer’s comment, we understand that the premise of this point needs further explanation. As a reminder, the calibration process started on the training data set where several model parameters in addition to lifestyle (shown in Table 1), including ones representing inherent physiological traits (e.g., effect of fatty acids on insulin signaling) were calibrated. These parameters, once calibrated, were transferred over to baseline-matched individuals in the test dataset. One of our hypotheses was that certain ranges or values or combinations of these physiological parameters could make weight loss easier, and that such trends would be observable through correlation of certain parameters with the response. In fact, similar hypotheses have been tested in clinical studies and corroborate what we found in our modeling analysis (e.g., this study by Gardner et al. https://pubmed.ncbi.nlm.nih.gov/26638192/). 

To answer the question about method, various methods including visual assessment, calculated correlation coefficients, and multiple linear regression were tested, and none revealed any significant correlations.

We have added some text in the paragraph to explain the reasoning behind this analysis (pg18-19, lines 412-423). 

--

10. P18/line 359: “no single optimal diet..”; do the authors mean that the optimal diet is not unique but different diets can achieve the same effect? It is stated, that 11 subjects did not see an improvement in the expected range (5-7% reduction in body weight). Can this be because of chance, where the sampled diet changes complemented the total caloric balance (e.g. small reduction in fat and carb but large increase in protein)?

The reviewer’s interpretation of the phrase “no single optimal diet” is correct. We have updated this sentence to remove ambiguity.

While the scenario described by the reviewer is theoretically possible, this is practically not the case because we sampled 2000 unique combinations of carbohydrate, fat, and protein changes, which adequately covers the sampled parameter space. We have added a new supplementary figure S9 that shows the distribution of the total caloric change due to the randomly sampled diets for 3 subjects. The distribution is normal with a mean of 0 and spans the ±25% range.

--

10.1 Regarding the experiment about analyzing hypothetical optimal diets (“Diet therapy is predicted to have maximal effectiveness when optimized individually”): More details about the experimental setup should be clarified and included in the Methods section. How many random simulations were carried out per individual? How was the MCMC set up? Why not simple latin-hypercube sampling (since there’s only 3 parameters being changed)?

We have added additional details (pg14-15, lines 322-328; pg19, lines 432-434) including the sample size and setup of the Monte Carlo simulations. We would like to point out that this was not a Markov Chain Monte Carlo (MCMC) simulation. 2000 sets of parameters were independently sampled and simulation performed independently. Since we were sampling only 3 parameters and had ample computational resources, economy was not a concern. Rather than limit sampling to certain types of combination, for instance by using latin-hypercube, we chose to obtain a large number of samples uniformly across the three dimensions. In the current context, this provides ample coverage of the parameter space.

--

11. P19/line 379-380: the 0.1-0.2% reduction in HbA1c is very close to the expected measurement error. In practice, this does not change the modelling result, but it should be mentioned.

We have added a sentence in Discussion to address this (pg27, lines 646-650).

--

12. In addition to the results already discussed on Fig 5. It is interesting to see the variation in “flexibility”. Some individuals seem to have a very narrow range of optimal diets. This also seems to correspond with how sensitive they are to carbs or fats, i.e. lines closer to a slope of -1 are longer (?).

The length of the lines in Figure 7 is limited by the searched space, which was in the range of –25 to +25% for both carbohydrates and fats. The lines are cutoff at these boundaries. Individuals in the bottom left corner appear to require rather large reductions in both carbohydrates and fats to achieve the target weight loss, and only a small number of diets in the sampled space fit that description. We have added text to explain the clipping of the lines at the parameter sampling boundaries (pg27, lines 640-646). 

--

13. P21/line 428: “however, tools that can enable customization of interventions at the individual level are lacking.” This may be very controversial since dietician advice can already be consider as customized to the individual. Maybe rephrase as “automated” or “model-based” customization or similar.

This is very good feedback. We have updated the sentence (pg23, lines 537-538).

--

14. P21/line 429-430: This is a very strong statement. See https://doi.org/10.21203/rs.2.20798/v1;
https://doi.org/10.1016/j.cell.2015.11.001; Of course, here the temporal scale is long term as opposed to short term (meal responses) and indeed the model may be argued to be mechanistic instead of fully data-driven. However, these distinctions should be made more explicit, as the general topic of using computational models to optimize diet is not new.

We thank the reviewer for pointing this out. We have updated the text to focus on the long-term mechanistic aspect of our work (pg23, lines 539-545).

Relevant References

1. Berry S, Drew D, Linenberg I, Wolf J, Hadjigeorgiou G, Davies R, et al. Personalised REsponses to DIetary Composition Trial (PREDICT): an intervention study to determine inter-individual differences in postprandial response to foods. Research Square; 2020. doi:10.21203/rs.2.20798/v1

2. Zeevi D, Korem T, Zmora N, Israeli D, Rothschild D, Weinberger A, et al. Personalized Nutrition by Prediction of Glycemic Responses. Cell. 2015;163: 1079–1094. doi:10.1016/j.cell.2015.11.001

3. Mazidi M, Valdes AM, Ordovas JM, Hall WL, Pujol JC, Wolf J, et al. Meal-induced inflammation: postprandial insights from the Personalised REsponses to DIetary Composition Trial (PREDICT) study in 1000 participants. Am J Clin Nutr. 2021;114: 1028–1038. doi:10.1093/ajcn/nqab132

--

15. P22/lines 444-458: instead of reiterating the experimental setup from them methods, perhaps include some discussion on how the choice of matching the training and test subjects may affect the interpretation of the results.

We appreciate the reviewer’s input and have updated the text accordingly (pg24-25, lines 570-581).

--

16. P23/line 470: It is stated that exogenous lifestyle factors were not fully predictive of the outcome. It is unclear what is meant by this. As far as I understand, the exogenous (lifestyle facotrs) parameters were the only thing changed in order to fit the response. Therefore, they must be predictive of the outcome, are they not? In addition, in this paragraph it is mentioned that the presented framework allows to capture the various interactions therefore it is well suited to understand the interactions. However, the study of the interactions is severely limited by the choice of fixed and varied parameters. From a biological perspective, on the scope of 4 years insulin resistance should also change in combination with some of the lifestyle parameters. This should be addressed. With more data, would it be possible to also estimate these parameters and truly uncover the interactions? What data would be ideal and in what quantity?

The statement about lifestyle not being fully predictive of outcome is about inter-individual variability and can be interpreted in a couple of different contexts based on the results. First, as shown in figure 4, similar caloric changes do not lead to the same amount of weight change for all individuals. Second, as presented in figures 5, 6 and 7, a wide range of diet changes can lead to similar changes in biomarkers (weight and HbA1c) and this spread is unique for each individual.

While the reviewer is right that only lifestyle parameters were altered in much of the analysis presented in this work, the fact that interindividual variability of response to lifestyle changes was captured by the model is due entirely to the variability in parameters other than the lifestyle factors, which were calibrated for the matched counterpart in the training dataset. The underlying details and the biological interactions represented in the model have been published previously (https://journals.plos.org/plosone/article?id=10.1371/journal.pone.0192472) and we chose not to delve into those details in this manuscript. We will, nonetheless, reiterate that differences between individuals were driven by non-linear interactions of the parameters other than lifestyle factors. Additionally, the structure of the differential equation system of the model represents biological interactions of the metabolic system in great detail. In fact, the reviewer’s comment about insulin resistance is naturally accommodated by the model. Insulin resistance is not represented by a single fixed parameter. Rather insulin signaling activity changes over time in response to changing levels of factors such as free fatty acids and reactive oxygen species (please see the “Insulin Resistance” section in the following article:

https://journals.plos.org/plosone/article?id=10.1371/journal.pone.0192472). 

We have briefly added some of these points and referred the reader to the original model for related details (pg26, lines 612-617). 

--

17. P25/line 517-520: these are more limitations (some are reiterated), move them to the appropriate paragraph about limitations.

The reviewer is right that the point about generalizability is already mentioned in the paragraph about limitations. We have deleted this sentence to avoid repetition.

--

18. Did the authors consider sharing an implementation of their model? Making it available through github (or equivalent) could increase the reach and impact of the authors’ work.

The data availability statement has reference to the simulation model which was used in the study. The PLOS ONE article which was referenced has the details of the model development that readers can benefit from.

--

19. Finally, please do a careful check for typos etc. The figure references seem to be not working and there are also some tracked changes left in the manuscript. Particularly, in the discussion there were many inconsistencies.

We have reviewed the text and references again for completeness and accuracy. 

--

 

Reviewer 2

There are one or two small errors in the text but none introduce ambiguity. eg Figure 5 x axis is misspelt. Please check the whole paper thoroughly. Otherwise it was a very well written paper.

This is a very well written paper. I found it difficult to read though because it is so information dense and the modelling language used is not very familiar to me, so it is my limitation as a referee of this type of paper that is the problem, and nothing wrong with the paper that I could see. I'm sure it will be understandable to a reader familiar with modelling in the health area. Because it is so well written, and the results are clearly presented I would be in favour of letting competent readers judge for themselves.

It seems that familiarity with previous J_DOIT1 would be required to fiully understand the paper.

The discussion is concise and the limitations of the study are well presented. The referencing seems to be thorough.

I responded "no" to question 3 because I though there should be a statement of data availability at the end of the paper.

We thank the reviewer for the positive feedback and for appreciating the relevance and presentation of the work. 

A statement about data availability was provided in the original submission and is available in the appropriate section. Briefly, all data are available without restriction. Individual patient level data can be requested from the authors and the simulation model has been previously published and is already in the public domain. A minimal dataset is shared that can be used to recreate the figures and main conclusions of the manuscript.

--

 

Editor

Dear editor, we ensured that the manuscript meets PLOS ONE’s style requirements, including file naming convention for figures.

--

This study focused entirely on analyzing data collected previously in the J-DOIT1 study. As such, no ethics statement is necessary for the current study.

--

"This work was supported by JSPS KAKENHI Grant Number 18k01988.The funders had no role in study design, data collection and analysis, decision to publish, or preparation of the manuscript. 

PricewaterhouseCoopers, LLP provided support in the form of salaries for the following authors - [JHC, MF, SP, PMD, SPV, GD] but did not have any additional role in the study design, data collection and analysis, decision to publish, or preparation of the manuscript." 

We included an amended financial disclosure statement within the cover letter.

--

The study’s minimal dataset is uploaded a supporting information file.

--

Included captions for each figure in the manuscript text in read order, below the paragraph where the figure is first cited.

--

---

## [Decision Letter · Decision Letter 1]

10 Oct 2023

PONE-D-23-14749R1Optimization of nutritional strategies using a mechanistic computational model in prediabetes: Application to the J-DOIT1 study dataPLOS ONE

Dear Dr. Sakane,

Thank you for submitting your manuscript to PLOS ONE. After careful consideration, we feel that it has merit but does not fully meet PLOS ONE’s publication criteria as it currently stands. 

We look forward to receiving your revised manuscript.

Kind regards,

Surya Prakash Bhatt, Ph.D

Academic Editor

PLOS ONE

Journal Requirements:

Reviewers' comments:

Reviewer's Responses to Questions

**Comments to the Author**

1. If the authors have adequately addressed your comments raised in a previous round of review and you feel that this manuscript is now acceptable for publication, you may indicate that here to bypass the “Comments to the Author” section, enter your conflict of interest statement in the “Confidential to Editor” section, and submit your "Accept" recommendation.

Reviewer #1: All comments have been addressed

Reviewer #2: All comments have been addressed

2. Is the manuscript technically sound, and do the data support the conclusions?

Reviewer #1: Yes

Reviewer #2: Yes

3. Has the statistical analysis been performed appropriately and rigorously? 

Reviewer #1: Yes

Reviewer #2: Yes

4. Have the authors made all data underlying the findings in their manuscript fully available?

Reviewer #1: Yes

Reviewer #2: Yes

5. Is the manuscript presented in an intelligible fashion and written in standard English?

Reviewer #1: Yes

Reviewer #2: Yes

6. Review Comments to the Author

Reviewer #1: I appreciate the significant effort made by the authors to improve the manuscript and to address my questions and comments. I recommend the manuscript for publication.

Reviewer #2: (No Response)

7. PLOS authors have the option to publish the peer review history of their article (what does this mean?). If published, this will include your full peer review and any attached files.

Reviewer #1: **Yes: **Balázs Erdős

Reviewer #2: No

---

## [Author Response · Author response to Decision Letter 1]

30 Oct 2023

In the previous review, the result was 'minor revisions,' but upon closer examination, it appears to be an 'acceptance.' I inquired about this matter, but there has been a change in the academic editor, and a new academic editor has taken over. I have not received a response regarding this issue. Therefore, we upload the revised manuscript without the highlights.　

Our diabetes model can simulate changes in body weight and glycemic control as a result of lifestyle interventions. In addition, this model offers the optimization of nutritional strategies through the utilization of a mechanistic computational model for prediabetes. While previous published work relied on genetic analysis or blood biomarkers, our model utilizes routinely collected laboratory data. We anticipate that our work will be of particular interest to diabetes professionals and healthcare providers involved in diabetes care.

---

## [Editor Report · Decision Letter 2]

3 Nov 2023

Optimization of nutritional strategies using a mechanistic computational model in prediabetes: Application to the J-DOIT1 study data

PONE-D-23-14749R2

Dear Dr. Sakane,

We’re pleased to inform you that your manuscript has been judged scientifically suitable for publication and will be formally accepted for publication once it meets all outstanding technical requirements.

Kind regards,

Surya Prakash Bhatt, Ph.D

Academic Editor

PLOS ONE
---

## [Editor Report · Acceptance letter]

21 Nov 2023

PONE-D-23-14749R2 

Optimization of nutritional strategies using a mechanistic computational model in prediabetes: Application to the J-DOIT1 study data 

Dear Dr. Sakane:

I'm pleased to inform you that your manuscript has been deemed suitable for publication in PLOS ONE. Congratulations! Your manuscript is now with our production department. 

Kind regards, 

on behalf of

Dr. Surya Prakash Bhatt 

Academic Editor

PLOS ONE